# Viral Vector Vaccines against Bluetongue Virus

**DOI:** 10.3390/microorganisms9010042

**Published:** 2020-12-25

**Authors:** Luis Jiménez-Cabello, Sergio Utrilla-Trigo, Eva Calvo-Pinilla, Sandra Moreno, Aitor Nogales, Javier Ortego, Alejandro Marín-López

**Affiliations:** 1Centro de Investigación en Sanidad Animal (CISA), Instituto Nacional de Investigación y Tecnología Agraria y Alimentaria (INIA), Valdeolmos, 28130 Madrid, Spain; luisfjim@ucm.es (L.J.-C.); sergio.utrilla@inia.es (S.U.-T.); calvo.eva@inia.es (E.C.-P.); sandramorenofdez@gmail.com (S.M.); nogales.aitor@inia.es (A.N.); 2Section of Infectious Diseases, Department of Internal Medicine, Yale University School of Medicine, New Haven, CT 06519, USA

**Keywords:** bluetongue virus (BTV), recombinant vaccines, viral vectors, humoral and cellular immune responses, multiserotype protection

## Abstract

Bluetongue virus (BTV), the prototype member of the genus *Orbivirus* (family *Reoviridae*), is the causative agent of an important livestock disease, bluetongue (BT), which is transmitted via biting midges of the genus *Culicoides.* To date, up to 29 serotypes of BTV have been described, which are classified as classical (BTV 1–24) or atypical (serotypes 25–27), and its distribution has been expanding since 1998, with important outbreaks in the Mediterranean Basin and devastating incursions in Northern and Western Europe. Classical vaccine approaches, such as live-attenuated and inactivated vaccines, have been used as prophylactic measures to control BT through the years. However, these vaccine approaches fail to address important matters like vaccine safety profile, effectiveness, induction of a cross-protective immune response among serotypes, and implementation of a DIVA (differentiation of infected from vaccinated animals) strategy. In this context, a wide range of recombinant vaccine prototypes against BTV, ranging from subunit vaccines to recombinant viral vector vaccines, have been investigated. This article offers a comprehensive outline of the live viral vectors used against BTV.

## 1. Introduction

Bluetongue virus (BTV) is a virus classified under the genus *Orbivirus*, within the family *Reoviridae*, and is transmitted via biting midges of the genus *Culicoides*. BTV is the causative agent of bluetongue (BT), a noncontagious arthropod-borne viral disease that affects both wild and domestic ruminants [1]. Certain breeds of sheep, especially fine-wool European breeds, and some species of wild ruminants, such as white-tailed deer, are the most commonly affected hosts, as they can show significant mortality rates [2,3], whereas cattle, goats, and the majority of wild ruminant species are usually asymptomatic. Nonetheless, cattle can be clinical upon infection (specially by BTV-8) [4] and, along with goats, can act as reservoirs for virus transmission from infected animals to other susceptible ruminants.

BTV virion is a non-enveloped icosahedral particle composed of three concentric protein capsid layers that surround a segmented genome [5,6]. Ten linear double-stranded RNA genome segments (S1 to S10) encode for seven structural (VP1–VP7) and five nonstructural proteins (NS1–NS5) [7,8]. The outer capsid layer contains two major proteins, VP2 and VP5, which are involved in cell attachment and membrane penetration, while the core is made up of the surface VP7 shell and the underlying VP3 layer [9]. Inside the core, there are transcriptase complexes formed by three minor enzymatic proteins, VP1, VP4, and VP6 [10,11]. The segmented nature of the BTV dsRNA genome enables the reassortment of genome segments when different serotypes or strains infect the host cell simultaneously [12,13], playing an important role in generating viral diversity. To date, 29 distinct serotypes of BTV, some of which are considered putative (serotypes 27–29) [14,15,16], have been identified all over the world [14,16], except in Antarctica.

BTV causes severe economic losses that are associated with its considerable impact on animal health, both direct such as weight loss, reduced fertility rate, reduced meat and milk production efficiency, and death, and indirect like lost revenue and trade restrictions [17,18]. To minimize these losses, vaccines have emerged as the most effective prophylactic measure to control BT disease and to potentially interrupt the cycle from the infected animal to the hematophagous vector. The focus of most current BTV vaccine research is on neutralizing antibody-based approaches; however, these are serotype specific. In fact, the specificity of interactions between BTV outer capsid proteins and neutralizing antibodies (Nabs) determines the identity of the BTV serotypes [19,20]. Cytotoxic T lymphocytes (CTLs) also play an important role in protective immunity against BTV; particularly, cell-mediated immune responses against nonstructural proteins are likely to be crucial in protecting against heterologous BTV serotypes [21,22,23,24]. However, antibody and CTL-based protection largely depends upon the nature of the vaccine platform applied. Typically, inactivated and subunit vaccines stimulate mainly antibody-based mechanisms, but they are poor stimulators of CTLs. On the other hand, live-attenuated and vectored vaccines may be potent inducers of both antibodies and CTLs [25]. Although inactivated vaccines are safer and can limit BTV dissemination, they cannot address the need for cross-protection among the different serotypes and do not allow for the distinction between infected and vaccinated animals (DIVA strategy). Live-attenuated vaccines (LAVs) have been widely used to control BTV in the past [26]. However, they are associated with teratogenicity, reversion to virulence, viremia that allows transmission to the insect vector, and risk of reassortment events with virulent wild-type viruses, giving rise to new virulent strains [27]. Recently, new strategies such as LAV based on reverse genetics [28,29] and viral vector vaccines have been designed to avoid these drawbacks. In this review, we aim to discuss the different prototypes of viral vectors designed to combat BTV, as well as their progress, strengths, and weaknesses, regarding multiserotype protection, safety, DIVA strategy, and long-term protection in both murine models and most important natural ruminant hosts.

## 2. Viral Vectors for Vaccine Applications

Viral vectors are regarded as potential tools for gene therapy and vaccine development. Their utility is predominantly based on the ability of viruses to infect cells, and the main advantages offered by viral vectors for vaccine development can be summarized as follows: (a) highly efficient gene transduction, (b) highly specific delivery of genes to target cells, (c) transient antigen expression, and (d) induction of robust immune responses, maintaining strong humoral immune responses and enhancing cellular immunity [30]. A successful presentation and delivery of antigens are crucial for inducing immunity and lifelong protection. Recombinant viral vectors have a potential for prophylactic use because they enable intracellular antigen expression and induce robust CTL response, leading to the removal of virus-infected cells. They are, therefore, ideal shuttles for delivering foreign proteins and also induce immune response by mimicking natural infection [30].

In addition, some attributes, such as the achievement of stable insertion of coding sequences into the genome, the aforementioned induction of a protective immune response, a proven safety record, and the potential for large-scale production, are required in order to qualify as a vaccine vector.

Multiple viruses have been used as vaccine viral vectors, ranging from very complex large DNA viruses such as poxviruses, down to simple RNA viruses such as parainfluenza viruses [31,32,33], where there are few restrictions imposed by gene packaging limits. Viral vector vaccines have been applied extensively in veterinary medicine. An outstanding example of this is Raboral V-RG (Merial), the first oral live vaccinia virus vector vaccine expressing the glycoprotein (GP) of Evelyn-Rokitnicki-Abelseth rabies virus [34,35].

In the following sections, we will describe the different viral vector-based vaccines that have been developed to prevent bluetongue disease, comprising viruses of different natures, such as poxvirus, adenovirus, herpesvirus, rhabdovirus, and phlebovirus.

## 3. Poxviruses

### 3.1. Vaccinia Virus and Modified Vaccinia Virus Ankara

Vaccinia viruses (VVs) have been engineered to express foreign genes, turning them into powerful vectors for recombinant protein expression. These were originated from highly efficacious vaccines for the eradication of smallpox [36], serving as a highly appealing delivery system for heterologous viral antigens [37]. The first approach to develop recombinant VV against BTV was described by Lobato et al., using the Western Reserve (WR) VV strain to construct recombinant VV expressing VP2 or VP5 of BTV-1, or coexpressing both BTV antigens [38] (Table 1). Notably, sheep immunized with the recombinant VV coexpressing VP2 and VP5 were able to develop high titers of Nabs, but lower in comparison with those sheep that received the LAV. Moreover, these two groups were not viremic, and animals did not display pyrexia following a challenge. Despite the non-negligible results of this work, the virulence of VV strains, particularly the WR strain, and the observation of a lower immunogenic profile compared with other highly attenuated VV strains entailed an insurmountable obstacle for their use as vaccine vectors [39].

To overcome these issues, researchers focused their efforts on the development of safer and immunogenic vectors. One of these highly attenuated strains, the modified vaccinia virus Ankara (MVA) strain, has been found to be immunogenic and useful for the application of protection against a vast number of infectious diseases [56]. Historical research focused on MVA and its use as vaccine against smallpox has allowed the scientific community to establish an extraordinary safety profile of this vector. This strain can be used under biosafety level 1 (BSL1) conditions because of its nature and its deficiency to productively grow in mammalian hosts. In addition, this replication-deficient viral vector has intrinsic capacities to induce both humoral and cellular immune responses. Historically, MVA was developed by serial tissue culture passage in primary chicken cells of vaccinia virus strain Ankara, and clinically used to avoid the undesirable side effects of conventional smallpox vaccination [57]. Adapted to grow in avian cells, MVA lost the ability to replicate in mammalian hosts and lacks many immunomodulatory genes that orthopoxviruses use to regulate the host cell environment [58,59,60]. Its ancestor virus is the vaccinia virus strain Ankara, which was originally propagated on the skin of calves and donkeys for smallpox vaccine production at the Turkish Vaccine Institute in Ankara. In 1953, the vaccinia virus strain Ankara was brought to Munich and added to the strain collection of the Institute for Infectious Diseases and Tropical Medicine at the University of Munich, where Herrlich and Mayr grew the virus on the chorioallantois membranes of embryonated chicken eggs and, therefore, named it as chorioallantois vaccinia virus Ankara (CVA) [61]. After serial passages in chicken (516th), it was renamed as modified vaccinia virus Ankara and given to the Bavarian State Institute for Vaccines to test its suitability for smallpox vaccine production [57,62].

Comparison with the genome maps of CVA ancestor viruses revealed that the MVA genome harbors six major deletions and mutations, resulting in the loss of ~30Kb of genetic material, which have altered virus–host interactions, as the absence of the A-type inclusion body protein or truncations in the HA promoter sequence [63].

The most common method used to produce recombinant MVAs involves the insertion of foreign genes into the thymidine kinase (TK) gene of the VV via homologous recombination [64]. This is accomplished through the construction of a recombination plasmid containing the gene of interest flanked by the VV TK gene. Afterwards, this plasmid is used to transfect previously MVA-infected cells, in which recombinant MVAs carrying the foreign gene are obtained by marker selection (Figure 1) [65].

The use of MVA as an expression vector for foreign genes was first described by Sutter and Moss, in which the expression of foreign reporter proteins such as LacZ was tested [66]. Since then, multiple recombinant MVAs have been generated against a plethora of human diseases [67,68,69,70,71,72]. The MVA vector has also been widely used against veterinary viral diseases [73,74,75,76]. For BTV (Table 1), the recombinant MVAs were first used for vaccination studies along with DNA vaccines, expressing the outer capsid proteins VP2 and VP5 from BTV-4. This regimen conferred a partial protection with reduced levels of viremia in the IFNAR(−/−) mouse model, using recombinant MVAs as boosters [40]. In the same work, the authors demonstrated the key role of the viral antigen VP7, observing a sterile protective effect in mice immunized with MVAs expressing these three antigens, suggesting the important role of Nabs induction (due to VP2) and the IFN-γ-secreting T cell activation (triggered by VP2 and VP7). Similar analysis was performed against BTV-8, a serotype with enhanced tropism for cattle that suddenly emerged in Northern-Central Europe in 2006 [77]. MVA–MVA or DNA–MVA prime-boost immunizations were performed, expressing VP2 alone, VP7 alone, or as a cocktail of MVAs expressing VP2, VP5, and VP7. The authors showed the capacity of VP2 in conferring protection against a lethal challenge of BTV-8 in the IFNAR(−/−) mouse model, whereas this level of protection was not achieved with VP7 alone [41].

As we already described, a myriad of different BTV serotypes have been reported. The approaches previously described have demonstrated good results regarding protection against homologous infections. However, an ideal vaccine against BTV would have to confer protection against multiple serotypes. To this end, researchers began to focus on those viral antigens that are more conserved among different serotypes and are able to induce a robust immune response in the host, when used as a vaccine candidate. Unlike VP2, which is the most genetically diverse antigen, the nonstructural protein NS1 is the most conserved protein among serotypes, and it has been described as a strong inducer of cellular immune response in both the mouse model and sheep [78]. Following this rationale, NS1 antigen was introduced in the vaccine composition along with VP2 and VP7, following a prime-boost immunization with DNA (prime) and MVAs (boost). The protective capacity of this strategy was evaluated in mice against the homologous challenge with BTV-4, showing sterile protection [22]. Subsequently, this strategy was also probed against heterologous infections with BTV-1 and BTV-8, showing 84% and 100% protection in immunized mice, respectively [22]. A Nab response was achieved (VP2 mediated), as well as a strong induction of the CD8+ T cell population, which was observed after stimulation of this cell subset with the three antigens used in the vaccine composition [22]. The broad protection observed against multiple BTV serotypes suggested that the protective role of NS1 and the cellular immune responses could be critical to achieve multiserotype protection.

Thereafter, the multiserotype protective role of NS1 was confirmed, observing that only this nonstructural protein vectorized in MVA in a homologous prime-boost immunization is necessary for conferring sterile protection against different BTV serotypes, like BTV-1, 4, 8, and 16, as well as the reassortant BTV-4 Morocco strain (BTV-4/MOR09) [23]. This study showed that mono- and multiserotype protection against BTV can be achieved in the complete absence of Nabs by enhancing cytotoxic CD8+ cellular immune responses. This work also showed that the protective capacity of NS1 resides in the N-terminal region (NS1-Nt), being dependent of a specific T cell epitope located in the amino acid position 152 (GQIVNPTFI) (peptide 152). The absence of this peptide in the NS1 amino acid sequence totally abrogates its protective ability [23].

MVAs have also been found to be protective in combination with other vaccine platforms, such as antigen presenting protein microspheres (µNS) carrying VP2, VP7, and NS1. In this case, this heterologous immunization strategy based on BTV-4 antigens was able to protect IFNAR(−/−) mice against serotypes 1 and 4 [42]. Moreover, MVAs have been combined with other viral vectors, like chimpanzee adenovirus Oxford 1 (ChadOx1), which will be discussed later [43]. In this study, the protective immune effect of MVA-NS1 was also evaluated in mice in a single-dose vaccination experiment, observing a delay in mortality and partial protection against a lethal challenge of BTV.

Another interesting approach was the generation of MVAs developed to combat viral infectious diseases that overlap in distribution or host and present a potential risk of expansion in nonendemic but close-to-endemic areas. This is the case of engineered recombinant MVAs against BTV and Rift Valley fever virus (RVFV). RVFV is a zoonosis that affects livestock, mainly sheep, and it is endemic in Africa and some regions in the Middle East. The appearance of outbreaks of RVFV in nonendemic areas like Europe is a potential threat, as there are different species of competent mosquito vectors, such as *Culex* and *Aedes*, already established in the area. Dual MVAs were generated in this study, cloning the GnGc gene of RVFV and the segments that encode VP2, NS1, and NS1-Nt from BTV in the *F13L* and *TK* loci, respectively, and under the control of VV early/late promoters. After a BTV challenge, all the immunized groups of IFNAR(−/−) mice showed protection, especially those immunized with NS1 and NS1-Nt, where 100% sterile protection was observed [44].

Finally, prompted by the high vaccination efficacy observed in the mouse model, the effectiveness of some of these promising candidates have been tested in the natural host. The dual MVA-GnGc-NS1 previously mentioned was tested against BTV-4 in sheep, using two doses of 10^8^ PFU per animal and observing very similar results in terms of rectal temperature and viremia. Additionally, vaccinated sheep were aviremic for an RVFV challenge (except one animal at day 3 postinfection), maintaining stable biochemical parameters (aspartate transaminase, gamma-glutamyltransferase, lactate dehydrogenase, and albumin), and had mild histological lesions compared with the nonvaccinated group, which indicated the bivalent character of the designed vaccine [44]. A similar trend was observed when MVA-NS1 was used as a booster of ChAdOx1-NS1 in a heterologous prime-boost immunization, as immunized sheep showed reduced levels of viremia and lower temperatures than the control group [43]. 

Although these results pave the way for the development of multiserotype vaccines against BTV in ruminants, further questions will need to be addressed, such as the exploration of other BTV viral antigens able to activate broad immune responses, the assessment of the long-term efficacy elicited by these candidates, and their capacity to reduce viremia that sufficiently avoids potential transmission by midge bites.

### 3.2. Other Poxviruses

Besides MVA, a variable set of other viruses belonging to the family *Poxviridae*, including capripoxviruses, avian poxviruses, and myxomaviruses, has been proposed as alternative vaccine platforms against BTV, but to a lesser extent (Table 1).

The genus *Capripoxvirus* (CaPV) comprises three closely related species (up to 97% nucleotide homology [79]) that are restricted to ruminant hosts: sheeppox virus (SPPV), goatpox virus (GTPV), and lumpy skin disease virus (LSDV). Attenuated capripoxviruses have been positively evaluated as vaccine vectors in ruminants [80,81,82,83], proving its safety and immunogenicity, and are considered ideal viral vectors because of their thermostability, large genome size, and ruminant host restriction, and because they are nonpathogenic to human hosts [84,85]. Interestingly, inoculation of these recombinant viral vectors induces a vector-specific immunity, which could eventually enhance the valence of the attenuated CaPV vaccine or even offer the possibility of constructing bivalent vaccines against both the viral vector used (CaPV) and the targeted viral agent [83,85]. Nonetheless, this pre-existing immunity may constrain their potential as vaccine vectors in ruminants, as it has been shown after the immunization of cattle with a recombinant CaPV encoding heterologous antigens from rinderpest virus [86]. Concerning BTV, a serotype cross-reactive, cell-mediated immunity was elicited in sheep by the recombinant live-attenuated strain KS-1 of LSDV expressing VP7 of BTV-1, observing partial protection against a heterotypic challenge with BTV-3 after a homologous prime-boost immunization regime [48]. BTV-specific ex vivo lymphocyte proliferation was also observed in goats after subcutaneous injection of a single dose of a recombinant capripoxvirus (KS-1 strain) individually expressing VP2, VP7, NS1, and NS3 of BTV-2 [49]. Nonstructural proteins of BTV are the predominant sources of antigens recognized by BTV-specific CD8+ CTLs [87], which have been described as critical for the development of a long-lasting immunity in animals infected with BTV [88,89]. However, mild protection was observed after a homotypic challenge in both sheep and goat, as animals displayed mild clinical signs but detectable levels of viremia after a challenge despite the inclusion of nonstructural proteins in vaccine design [49].

To date, two avipoxviruses have been exploited as viral vectors against BTV: fowlpox (FPV) and canarypox viruses (CPV). Recombinant FPV and CPV vaccines expressing foreign antigens have been proved safe and effective in mammalian hosts [90,91,92,93,94,95,96,97,98,99,100]. In addition to having a large cargo capacity of both viral vectors, these exhibit an ideal safety profile due to their natural host range restriction to avian species and abortive replication in mammalian and insect cells, which makes them a safer but effective alternative to other live virus vectors [101,102,103,104]. For BTV, a recombinant FPV coexpressing genes encoding the VP2 and VP5 outer capsid proteins of BTV-1 administered in combination with a DNA vaccine prime elicited humoral and BTV-specific T-cell responses in BALB/c mice and significant and sustained levels of serum Nabs in sheep [45]. However, the protective capability against BTV was not analyzed further. Regarding CPV, serotype-specific protection was observed in sheep subjected to a homologous prime-boost vaccination regime with a recombinant CPV coexpressing VP2 and VP5 proteins of BTV-17, as BTV particles were not isolated from the blood of vaccinated sheep after challenge [46].

*Myxomavirus* (MYXV), a leporide-specific poxvirus, is also a potential nonreplicative vector for ruminant immunization. Like FPV and CPV, MYXV abortively infects ruminant cells, allowing the expression of substantial amounts of foreign genetic material [105,106]. Furthermore, this viral vector has shown vaccine efficacy and safety in several mammalian hosts, including sheep [106,107,108]. With regard to BTV, VP2, individually or in combination with VP5, was vectorized in a recombinant MYXV. After a homologous prime-boost immunization regime with the recombinant MYXV expressing VP2 alone, immunized sheep displayed higher levels of viremia and more severe clinical signs than sheep vaccinated with inactivated BTV-8, but animals were better protected from a homotypic BTV-8 viral challenge than the non-immunized control group [47]. It has been described that VP5 enhances the protective immune response elicited by VP2 alone [109,110]. Conversely, the protection conferred by the recombinant MYXV simultaneously expressing VP2 and VP5 was similar to that of the negative control group, which could rely on a diminished expression of VP2 by the recombinant vector and/or nuclear localization of VP5 observed, which could mismatch its likely conformational influence on VP2 (mainly located in the cytoplasm) as pointed out by the authors, thus impairing the induction of a potent humoral immune response. The immunogenicity of another recombinant MYXV (derived from the attenuated vaccine strain of MYXV SG33) expressing the VP7 protein of BTV-2 has also been assessed, observing an induction of a VP7-specific CD4+ cell subset [52].

In summary, these poxviral vectors are a potential alternative to inactivated vaccines or LAVs as they are immunogenic and naturally attenuated in ruminants. Moreover, these reviewed strategies could be further used in combination with MVA to evade vector-specific immune responses, thus boosting the immunogenicity of recombinant MVAs. Nonetheless, further research is needed to improve the efficacy of this poxviral vector in ruminant species, which will be important not only for BTV but also for other relevant virulent veterinary diseases.

## 4. Herpesviruses

The large genome size and the dispensability of virulence genes for productive viral replication in vitro and in vivo are some of the advantages of using herpesviruses as vaccine vectors. Bovine (BoHV) and equine herpes (EHV) viruses have been used to construct recombinant vectored vaccines against BTV (Table 1).

The genome of a nonpathogenic strain of BoHV-4, which belongs to the *Gammaherpesvirinae* subfamily, was cloned as a bacterial artificial chromosome (BAC) and manipulated to be used as a vector platform to deliver chimeric peptides (Figure 2) [111]. It is known that the cellular location of an antigen may alter the level of immune response elicited [112]. Membrane-anchored antigens usually induce a stronger immune response than secreted forms of the same antigen [111]. In this sense, a BoHV-4-based vector was engineered to express the BTV-8 immunodominant VP2 provided with a heterologous signal peptide to its amino-terminal and a transmembrane domain to its carboxyl-terminal region (IgK-VP2gDtm), thus allowing the VP2 expression targeting to the cell membrane fraction. Mice vaccinated with recombinant BoHV-4 expressing VP2 developed serum Nabs and a reduction in viremia following BTV-8 virus challenge. All immunized animals showed a delay in clinical onset and death, with a single mouse showing no viremia and surviving [50]. Therefore, BoHV-4-VP2 elicited partial protection against homologous BTV-8 in these experiments. In a different approach, EHV-1, of the *Alphaherpesvirinae* subfamily, was used as the delivery vaccine vector for BTV. EHV-1 strain RacH, attenuated by continuous cell passaging, has been established as an infectious BAC that facilitates the manipulation of the virus genome [113]. The outer capsid protein VP2 of BTV-8 (rH-VP2), alone or combined with VP5 (rH-VP2-VP5), was vectorized in two EHV-1 RacH-based recombinant vaccines and evaluated in the mouse model [51]. Immunization with rH-VP2 did not protect animals after challenge with BTV-8, as all vaccinated mice died. However, recombinant EHV-1 carrying both VP2 and VP5 (rH-VP2-VP5) elicited substantial protection against BTV-8 infection. Mice only transiently displayed mild disease, but no infectious virus was found, and the mice fully recovered.

## 5. Adenoviruses

Adenoviruses (Ads) have transitioned from tools for gene therapy to bona fide vaccine delivery vehicles against numerous infectious diseases. Additionally, they are being explored against a multitude of tumor-associated antigens due to the strong innate and adaptive immune responses that this vector elicits in mammalian hosts [114,115].

Ads show a large natural diversity and a broad spectrum of hosts. However, replication-competent and replication-deficient Ad vectors with therapeutic applications have been built mainly around human Ad type 5 (Ad5) [116]. Although human Ad5 has been widely used in gene therapy, it is also being studied as a vaccine candidate against different infectious diseases, such as the simian/human immunodeficiency virus [117], Zika virus [118], and Ebola virus [119,120,121,122]. Indeed, its efficacy as a vaccine platform is also being tested against severe acute respiratory syndrome coronavirus 2 (SARS-CoV-2) [123,124,125]. Similar approaches have been performed using simian Ads, which include replication-defective chimpanzee adenoviruses (ChAd). These vectors are safe and induce strong cellular and humoral immunity, and also lack problems related with pre-existing antivector immunity [126]. Vaccine vectors based on recombinant ChAdOx1 (a ChAd viral vector developed by Oxford University) have also been tested with promising results in many animal models for numerous infectious diseases [127,128,129,130,131,132,133,134,135,136,137,138,139,140]. They have also been used against diseases of veterinary importance like RVFV with great success in sheep, goats, and cattle (target hosts of RVFV), which were totally protected against a viral challenge after immunization with ChAdOx1-GnGc vaccine [141]. 

Ad vectors have been used against BTV (Table 1). For instance, canine adenovirus type 2 (CAV-2) has been tested against BTV [52]. This nonreplicative vector bypasses problems related to pre-existing antivector immunity and has demonstrated its potential as an antigen delivery platform [142,143]. CAV-2 expressing VP7 of BTV-2 was successful in inducing both CD4+ and CD8+ T cell responses in sheep but failed to confer protection against the homologous (BTV-2) and heterologous (BTV-8) challenges [52]. 

Replication-defective rAd5 expressing either VP7, VP2, or NS3 BTV-8 proteins was used to immunize both IFNAR(−/−) mice and sheep (combining rAd5-VP7, rAd5-VP2, and rAd5-NS3 or either rAd5-VP7 or rAd5-VP2 alone). Mice were protected against a homologous BTV challenge, although viremia was not determined and developed humoral and BTV-specific CD8+ and CD4+ T cell responses after vaccination. Sheep vaccinated with rAd5-VP2 + Ad5-VP7 or only with rAd5-VP7 and challenged with BTV showed mild disease signs and reduced pyrexia and viremia levels. As expected, rAd5-VP2 + rAd5-VP7 induced Nabs (directed against VP2). Remarkably, the partial protection conferred by rAd5-VP7 was achieved in the absence of Nabs, by the induction of strong BTV-specific CD8+ T cell responses, suggesting an important role of the T cell immunity mediated by VP7 in protection against BTV infection [53].

In the same line, ChAdOx1 vectors expressing the viral NS1 protein, or its truncated form NS1-Nt, were generated and assessed as vaccine candidates in a single-shot immunization and in a heterologous prime-boost regimen combined with recombinant MVAs expressing the same antigens. A single dose of ChAdOx1-NS1 or ChAdOx1-NS1-Nt induced a moderate CD8+ T cell response and protected IFNAR(−/−) mice against a lethal dose of BTV-4/MOR09 (total for ChAdOx1-NS1 and partial for ChAdOx1-NS1-Nt), showing reduced levels of viremia after infection compared with the control group. Interestingly, IFNAR(−/−) mice immunized with a single dose of ChAdOx1-NS1 were totally protected after a BTV-8 challenge, displaying undetectable levels of viremia and absence of clinical signs. Additionally, the heterologous prime-boost ChAdOx1/MVA expressing NS1 or NS1-Nt elicited a robust NS1-specific cytotoxic CD8+ T cell response and protected the animals against BTV-4/MOR09 even 16 weeks after immunization, with undetectable levels of viremia at any time after a challenge. Subsequently, the best immunization strategy, based on ChAdOx1/MVA-NS1, was assayed in sheep. Despite viremia being detected in immunized sheep, the level of virus in the blood was 100 times lower than in non-immunized animals, which also presented clinical signs and pyrexia [43]. This study reaffirmed the important role of the NS1 antigen in eliciting a multiserotype protective effect mediated by the activation of the cellular cytotoxic CD8+ responses and showed the strong capacity of ChAdOx1 as a vector vaccine, inducing protection and CD8+ T cell activation just with a single shot. Finally, the use of this vector in a heterologous prime-boost immunization with the MVA vector was shown to be sufficient and efficient for conferring long-lasting protection, one of the main challenges in vaccine design.

## 6. Other Viruses

Vesicular stomatitis virus (VSV) has been widely employed as a recombinant VSV vector against a number of different pathogens [144,145,146,147,148,149,150] because of the strong induction of the humoral immune response triggered by this virus [147]. For the generation of recombinant VSV vectors, VSV G protein is replaced by the antigen of interest. A helper cell line providing the VSV G protein in trans is used for the generation of recombinant VSV replicon vectors (Figure 3a). VSV replicons infect a broad spectrum of cell types, but they are unable to complete a whole replication cycle due to the deletion of the G protein gene, even though recombinant genes are expressed at high levels. A VSV vector based on BTV antigens was developed by Kochinger et al. (Table 1). Propagation-incompetent VSVΔG vectors expressing the BTV-8 outer capsid proteins VP2 or VP5, or a combination of both, were used for sheep immunizations, following a prime-boost strategy. As expected, serum Nabs were induced in those animals immunized with VP2. Afterwards, immunized sheep were infected with BTV-8. VSV-VP2- or VSV-VP2-VP5-immunized animals were protected in the absence of clinical signs, viremia, and pyrexia, whereas VSV-VP5 did not confer a protective effect [54].

A novel approach has recently been developed by our lab, in which RVFV has been used as a delivery viral vector for BTV antigens (Table 1). RVFV is a *Phlebovirus* with a three-RNA-segmented genome of negative and ambisense polarity [151]. The S (small) segment encodes in an ambisense orientation the viral nucleoprotein N and the nonstructural protein NSs. The latter is considered a virulence factor responsible for host general transcription suppression, suppression of antiviral IFN-β, and the facilitation of viral antigen translation. Interestingly, natural NSs deletion mutants have been found, presenting an attenuated or avirulent phenotype [152], making them suitable for their use as a vaccine candidate. Furthermore, the strong irruption of reverse genetics techniques has facilitated the generation of RVFV with modified genomes lacking nonstructural genes [153], allowing the generation of attenuated RVFVs’ viral vectors that encode and express heterologous genes [154]. Recombinant RVFVs able to express either a truncated BTV-4 NS1 protein, NS1-Nt, or the VP2 capsid antigen were generated (RVFV-NS1-Nt and RVFV-VP2), substituting the NSs gene for the heterologous BTV genes (Figure 3b). Their stability was tested and characterized phenotypically prior to their use as immunogens. Their ability to induce protective immune responses was tested in mice and sheep as well [55]. In both animal models, the VP2 expression level was enough to elicit Nab levels. Nonetheless, sheep vaccinated with RVFV-NS1-Nt were better protected against BTV as determined by the levels of viremia and lower anti-BTV VP7 antibody levels (indicative of lower BTV replication in sheep) after a challenge. This, together with the fact that BALB/c mice immunized with RVFV-NS1-Nt and then boosted with BTV-4 developed an NS1 epitope-specific CD8+ T cell response, may contribute to support the use of a rRVFV-BTV-4 vaccine vector for BTV [55].

## 7. Conclusions and Future Perspectives

The prevention of BTV infections has been a challenging task due to uncontrolled insect vector expansion, the number of circulating serotypes of BTV in combination with the absence of strong cross-protection among them, and the ability of BTV to reassort [155,156,157]. In addition, the impact of BTV outbreaks is greater due to the risk of epidemics in regions where BTV was not circulating before. Although vaccination is the most cost-effective strategy to control and prevent BTV infections, and allows for cushioning the economic impact of BT disease in nonendemic areas, the effectiveness of currently marketed BTV vaccines is suboptimal [40,43,158,159,160].

New-generation vaccine strategies against BTV, including universal or multiserotype vaccines, must induce more robust and long-lasting and cross-protective immunity, and clearly maintain a high safety profile. The implementation of viral vector vaccine approaches to prevent BTV infections has provided the opportunity to develop more effective prophylactic strategies to protect animals against BTV infections. Moreover, the differentiation of infected from vaccinated animals (DIVA), now possible with this new generation of vaccine approaches, could eventually ameliorate the economic losses caused by BTV.

Although production and regulatory concerns have prolonged vaccine rollout for veterinary use, the looming threat of BTV outbreaks affecting the food chain warrants all options to limit virus spread. There are several advantages and disadvantages associated with the described viral vector vaccine approaches against BTV, although all systems have been demonstrated to be safe and efficacious experimentally. Importantly, attention should be paid to the genetic stability of these new vaccine candidates. In addition, more large-scale field trials need to be performed, and all systems should be compared side by side to determine which approaches are superior for the successful generation of BTV vaccines. However, the abundant safety, economic, and biological advantages of these technologies highlight their promising future to control and even partially eradicate BTV.

## Figures and Tables

**Figure 1 microorganisms-09-00042-f001:**
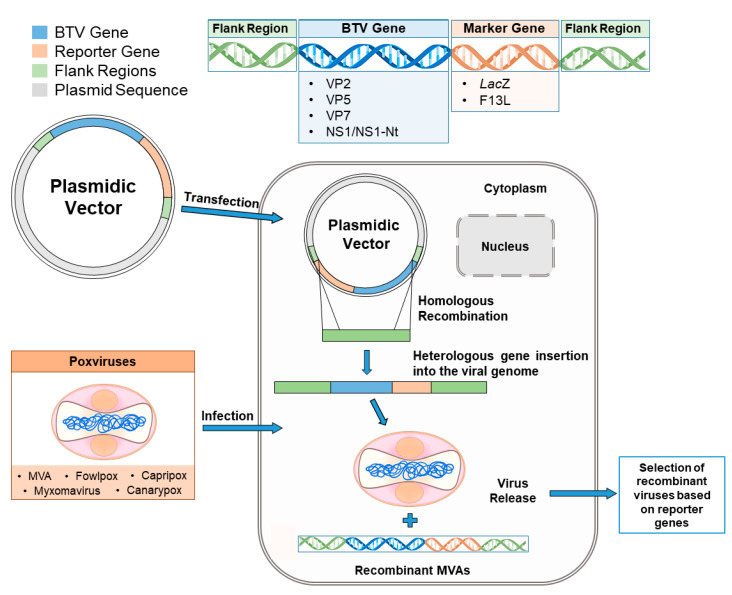
Diagrammatic representation of one example of cloning strategy for the generation of recombinant poxviral vector. A homologous recombination process takes place in the cytoplasm of eukaryotic cells simultaneously infected with the virus vector and transfected with a plasmid vector encoding the BTV gene of interest (blue) along with a marker gene (orange). Flank regions (green) allow for recombination and heterologous gene insertion into the viral genome. The BTV genes cloned and the marker genes used in several works are listed above. Finally, after a selection process based on the marker gene, the recombinant poxvirus expressing BTV genes is recovered.

**Figure 2 microorganisms-09-00042-f002:**
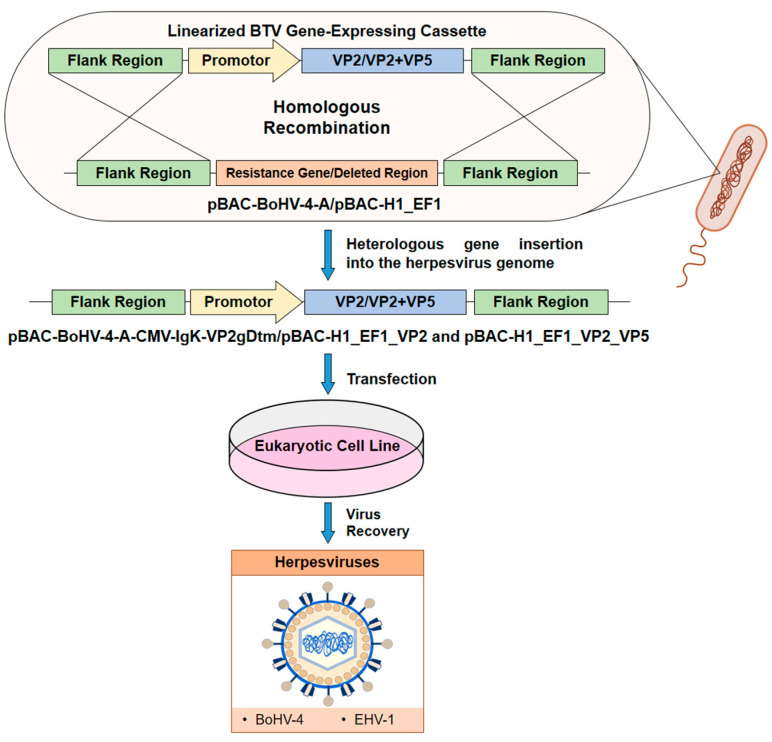
Schematic representation of the generation procedure to obtain recombinant herpesviral vector. A plasmid containing the BTV gene of interest (blue) flanked by homologous regions (green) is linearized and used to transform competent bacteria. Simultaneously, a bacterial artificial chromosome (BAC) encoding the genome of the herpesvirus is also used to transform these prokaryotic cells. Homologous recombination between flank regions allows for obtaining a pBAC encoding the BTV gene. After transfection of this pBAC containing the BTV gene of interest in a eukaryotic cell line, recombinant herpesviruses are recovered.

**Figure 3 microorganisms-09-00042-f003:**
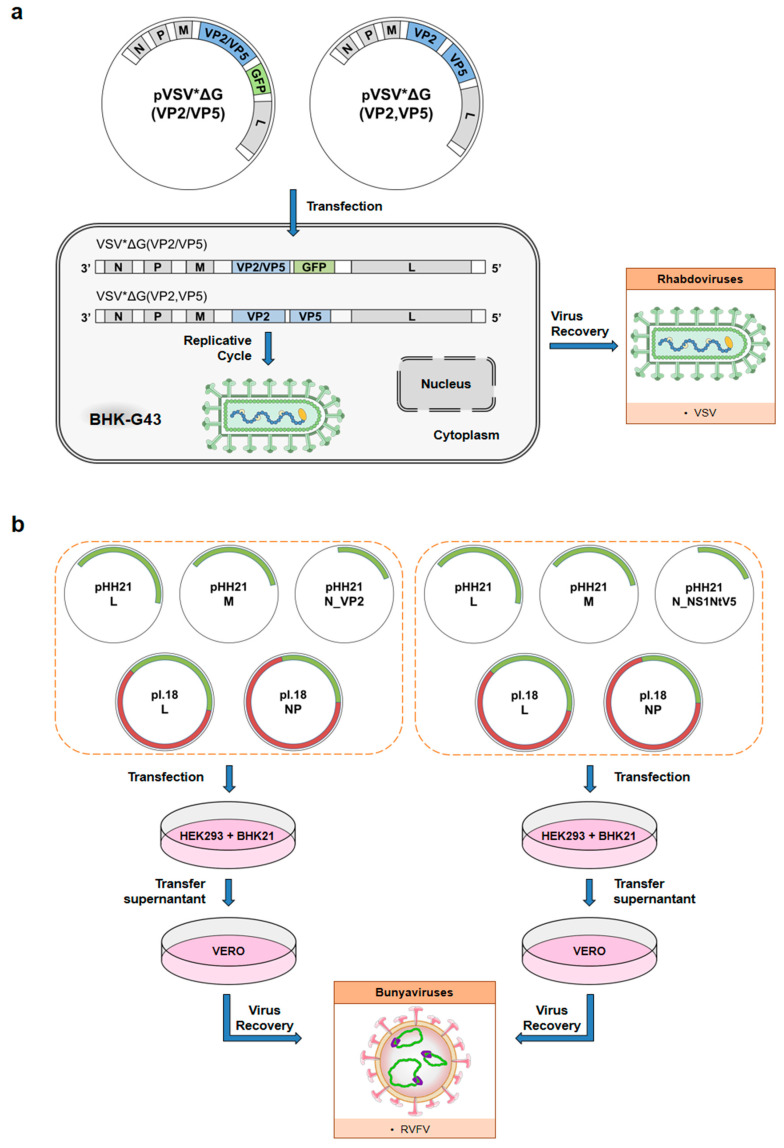
Overview of the generation procedure of recombinant RVFV and VSV. (**a**) Plasmid pVSV* was used to construct plasmids pVSV*ΔG(VP2) and pVSV*ΔG(VP5) by replacing the VSV G gene with the BTV gene of interest. Plasmid pVSV*ΔG(VP2,VP5) was constructed from plasmid pVSV*ΔG(VP2) by incorporating an additional transcription unit encoding the VP5 protein of BTV. After the transfection of each plasmid in a helper cell line providing the VSV G protein in trans (BHK-G43), VSV replicon particles (VRPs) were generated and recombinant VSV expressing BTV genes was recovered. (**b**) Rescue of recombinant RVFV is conducted with the pol I/II rescue system in HEK293 and BHK-21. A co-culture of HEK293 and BHK-21 is transfected with a combination of five plasmids, three of them carrying the viral genome with the heterologous antigen (pHH21) and the remaining two expressing helper proteins (pL18). The supernatants of transfected cells are harvested on days 3, 5, and 7 post-transfection, and the presence of recombinant virus is screened in Vero cells by means of cytopathic effect (CPE) detection or immunofluorescence with specific antibodies.

**Table 1 microorganisms-09-00042-t001:** Overview of viral vector vaccine candidates against BTV.

Viral Vector	Type of Virus	Antigen Included	Type of Induced Immune Response	Animal Model	Multiserotype Protection	Safety	Ref.
VV Western Reserve strain	dsDNA	VP2/VP5/VP2 + VP5	Humoral	Merino sheep	In vitro cross-neutralization	Virulent strain	[38]
Modified vaccinia virus Ankara (MVA)	dsDNA	VP2, VP5, VP7, and NS1	Humoral and cellular	IFNAR(−/−) mice Churra sheep	Yes (Not tested in sheep)	Safe	[22,23,40,41,42,43,44]
Fowlpox virus (FPV)	dsDNA	VP2 + VP5	Humoral	BALB/c mice Sheep (breed not specified)	No	Safe	[45]
Canarypox virus (CPV)	dsDNA	VP2 + VP5	Humoral	Dorset sheep	No	Safe	[46]
Myxomavirus (MYXV)	dsDNA	VP2/VP2 + VP5	Humoral	Lacaune lambs	No	Safe	[47]
Capripoxvirus (CaPV)	dsDNA	VP7	Cellular	Dorset sheep	Partial	Safe	[48]
		VP2 + VP7 + NS1 + NS3	Humoral and Cellular	Saanen goats and Préalpes sheep	Presumable (not tested)	Safe	[49]
Bovine herpesvirus 4 (BoHV-4)	dsDNA	VP2	Humoral	IFNAR(−/−) mice	No	Safe	[50]
Equine herpesvirus 1 (EHV-1)	dsDNA	VP2 + VP5	Humoral	IFNAR(−/−) mice	No	Safe	[51]
Canine adenovirus type 2 (CAV-2)	dsDNA	VP7	Cellular	Préalpes sheep	No	Safe	[52]
Adenovirus type 5 (Ad5)	dsDNA	VP7, VP2, and NS3 (different combinations)	Humoral and cellular	IFNAR(−/−) mice Colmenareña sheep	Presumable (not tested)	Safe	[53]
Chimpanzee adenovirus 1 (ChAdOx1)	dsDNA	NS1/NS1-Nt	Cellular	IFNAR(−/−) mice Churra sheep	Yes	Safe	[43]
Vesicular stomatitis virus (VSV)	ssRNA(−)	VP2/VP5/VP2 + VP5	Humoral	Swiss White Alp sheep	No	Safe	[54]
Rift Valley fever virus (RVFV)	ssRNA(−)	VP2/NS1-Nt	Humoral and cellular	BALB/c mice Churra sheep	Presumable (not tested)	Safe	[55]

dsDNA: double-stranded DNA; ssRNA(−): negative-sense single-stranded RNA

## Data Availability

Not applicable

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
