# Peer review of "Viral Vector Vaccines against Bluetongue Virus"

_microorganisms, 2020, doi:10.3390/microorganisms9010042_

Round 1

Reviewer 1 Report

The review paper entitled “Viral vector vaccines against bluetongue virus” reports the knowledge on the current design of vector based vaccines for BTV.

The main challenge in BTV vaccination is the conventional approach to elicit humoral immunity but there has been a movement towards generating cellular immunity. The authors have briefly described both types of immunity in the introduction and in various portions of the review reported on vaccination strategies to stimulate either types of immunity. The paper would benefit from additional introduction of reviewing which virus proteins are key to humoral or cellular-mediated immunity and to link that further to how certain viral vectors are better for cellular immunity, which the authors have alluded that NS1 vaccines in their pipeline might work better. On the same note, I would discourage authors to include their preliminary findings of their work as it is not peer-reviewed and is not appropriate to be part of a review paper.

This paper has also identified the key gap of creating a broad serotypic vaccination whilst mentioning serotype specific NS protein vaccination strategy. The authors have recognised that some NS proteins are highly conserved and should have also understood that these proteins are not responsible for serotypic-specific responses. This brings to the same point mentioned above that discussions should be directed towards how immunity is generated for each NS and how specific viral vectors modulate or fail to provide immunogenic stimulation. Despite murine models are convenient models for vaccine evaluation, the authors should also describe in details how the immunity differs between murine and ovine and hence the inconsistency of some vaccine results between the two species.

Further editorial comments are provided in the PDF.

Author Response

We would like to thank the reviewer for the comments on our manuscript that will help to improve the manuscript. The preliminary findings of the original manuscript as it is not peer-reviewed and the figure 2 have been deleted as suggested by the reviewer. In addition, information about which virus proteins are key to humoral or cellular-mediated immunity has been included and how the experimental vaccines elicit the immune response and protection in mice and sheep has been specified. We have carefully modified the points or include the information requested by the reviewer throughout the manuscript and the manuscript has been modified accordingly in the new version.

Reviewer 2 Report

Very good review. A few comments

  • in the adenovirus part some aspects of "failure" with adenovirus vectors should be described (the authors should talk about Bouet-Cararo et al, 2014)
  • line 59, please reconsider the sentence: I'm not ok with the "potentially" because you cite an example of reassortment. The sentence should be corrected

Author Response

We would like to thank reviewers for the positive comments on our manuscript.

The reference Bouet-Cararo et al, (2014) has been included in the manuscript in the adenovirus section.

In line 59, the sentence has been corrected by removing the word " potentially¨.

Round 2

Reviewer 1 Report

The authors have addressed issues that have been brought up previously. There is only minor editorial comments in this review. Please see attached PDF for comments. 

Author Response

We have carefully answered the points made by the reviewer and the manuscript has been modified accordingly.